# Trends in Cigarette Smoking among Middle-Aged Lithuanian Subjects Participating in the Primary Prevention Program between 2009 and 2016

**DOI:** 10.3390/medicina55050130

**Published:** 2019-05-12

**Authors:** Egidija Rinkūnienė, Žaneta Petrulionienė, Vilma Dženkevičiūtė, Silvija Gimžauskaitė, Antanas Mainelis, Roma Puronaitė, Agnė Jucevičienė, Urtė Gargalskaitė, Aleksandras Laucevičius

**Affiliations:** 1Department of Cardiovascular Medicine, Vilnius University, 08661 Vilnius, Lithuania; zaneta.petrulioniene@santa.lt (Ž.P.); agne.juceviciene@santa.lt (A.J.); urte.gargalskaite@santa.lt (U.G.); cardio@santa.lt (A.L.); 2Faculty of Medicine, Vilnius University, 08661 Vilnius, Lithuania; vilma.dzenkeviciute@santa.lt (V.D.); s.gimzauskaite@gmail.com (S.G.); 3Centre of Cardiology and Angiology, Vilnius University Hospital Santaros Klinikos, Santariškių Str. 2, 08661 Vilnius, Lithuania; 4Vilnius University Hospital Santaros Klinikos, 08611 Vilnius, Lithuania; antanas.mainelis@santa.lt (A.M.); roma.puronaite@santa.lt (R.P.)

**Keywords:** smoking prevalence, smoking cessation, primary prevention, risk factors, trends

## Abstract

*Background and Objectives*: The aim of the study was to evaluate trends in smoking among middle-aged men and women based on the data from the Lithuanian High Cardiovascular Risk (LitHiR) primary prevention program between 2009 and 2016. *Materials and Methods*: A community-based cross-sectional study comprised 92373 Lithuanian adults (41.6% men and 58.4% women). We compared the prevalence of smoking, smoking cessation activity and smoking intensity by gender and age groups. *Results*: There was a statistically significant upward trend in the number of smoking subjects (average annual percent change (AAPC) 2.99%, *p* < 0.001). The number of smoking male subjects remained much higher than the number of smoking female subjects during the 2009–2016 period. The study showed a significant increase in the percentage of smoking quitters in the whole group (AAPC 7.22%, *p* < 0.001) and among men and women separately. There was no significant change in smoking intensity in groups of male and female smokers separately. *Conclusions*: The analysis showed that the prevalence of smoking in Lithuania is still increasing due to women smoking despite all the governmental tobacco control efforts to reduce it.

## 1. Introduction

Every year, cardiovascular disease (CVD) accounts for more than 17 million premature deaths globally, and it is likely to reach 23.6 million by the end of 2030 [1,2]. It is also responsible for 3.9 million yearly deaths in Europe and over 1.8 million deaths in the European Union [3]. CVD causes twice as many deaths as is caused by cancer and more than any other remaining disorders combined. Although the mortality rate from CVD is decreasing in most European countries, including Central and Eastern Europe, it is still generally higher in Lithuania than in Northern, Southern and Western European countries [4]. According to the data of The Institute of Hygiene, cardiovascular disease remains the leading cause of death in Lithuania (48.1% among men and 63.4% among women deaths) [5]. The most common causes of death from CVD events in Lithuania are ischemic heart disease and stroke [6].

Cardiovascular disease is caused by many health determinants including biological, behavioral, psychological, socio-economic and environmental factors [1]. Smoking is the leading avoidable cause of CVD and premature death worldwide [7]. Tobacco use is estimated to cause about 10% of CVD, and it is the second leading cause after high blood pressure [8]; 17.7% deaths caused by ischemic heart disease and 18.6% deaths caused by stroke are related to tobacco use. In 2016, smoking caused 21.62% among male deaths and 4.38% among female deaths in Lithuania [9].

The World Health Organisation (WHO) has reported that smoking prevalence in 2017 was 28% in Europe being the highest rate compared among all six WHO regions [10]. According to Eurostat, the highest numbers of daily smokers were estimated between the ages of 25 and 54, considerably decreasing in the age groups of 65 and over [11]. It indicates the relevance of considering before-mentioned age range as long as CVD usually manifests in middle-aged people [12].

In clinical practice, smoking is often the overlooked determinant of CVD, receiving much less attention than hypertension, dyslipidemia or diabetes. Smoking cessation determines a significant reduction in CVD risk. Quitting smoking before the age of 40 reduces the risk of tobacco-related death by 90%. Quitting at any age dramatically reduces risk of myocardial infarction and stroke, and excess risk of CVD eliminates within 2 years [13]. The WHO has calculated that 1 year after smoking cessation, the risk of CVD reduces to half that of a smoker’s and after 15 years reaches the same risk as a nonsmoker’s [14]. There are essential long-term health benefits, and smoking should be treated as actively as other CVD risk factors [13].

## 2. Materials and Methods

The Lithuanian High Cardiovascular Risk (LitHiR) primary prevention program is applied to evaluate risk factors of cardiovascular disease in middle-aged men and women. The program is funded by the Ministry of Health and is available for men aged 40–54 and women aged 50–64 without overt cardiovascular disease. The number of primary health care institutions participating in the LitHiR program was 398/420 (94.8% of all the primary care institutions in Lithuania), which uniformly covers the whole country. In primary health care institutions, special questionnaires were completed. This report describes the trends in smoking in the randomly selected group of 92,373 subjects involved in the LitHiR programme during the period 2009–2016 at the primary level. In 2016, 256,625 adults were examined in primary care centers, covering about 37.5% of all target population. Considering the number of participating primary health care institutions and the number of participants examined, we believe that our study reflects the general population of middle-aged Lithuanian people. The study protocol was approved by Vilnius Regional Biomedical Research Ethics Committee (No. 158200-15-816-329).

There were three ways subjects entered the LitHiR programme: by suggesting to proper age patients who came to the primary health center for any reason to participate in the program; by actively inviting people who fit the criteria of the program after examinating their medical history; and by promoting the program via mass media. Subjects underwent analysis of antrhopometric measurements (height, weight, waist circumference, body mass index), physical examination (heart rate and blood pressure), and evaluation of presence or absence of associated cardiovascular risk factors (smoking, physical activity, dietary habits). The overall cardiovascular risk was estimated using Systematic Coronary Risk Evaluation (SCORE) system. More detailed information on LitHiR study protocol can be found in another article [15].

Despite other existing types of tobacco use, only cigarette smoking was analysed as one of the risk factors for cardiovascular disease in this article. All groups of gender and age were compared by the prevalence of smoking, smoking cessation activity and smoking intensity. Smoking intensity was evaluated by asking the number of cigarettes usually smoked per day and results were categorised into groups of <10, 10–20 and >20 cigarettes per day. Quitting smoking was determined when a subject answered that he or she was smoking earlier in their life.

### Statistical Analysis

SPSS 17.0 statistical software was used for data analysis. Arithmetic mean, standard deviation (SD) and 95% confidence interval (CI) were calculated for quantitative variables. Categorical data were summarized by frequencies. Chi-squared test was performed to compare categorical variables. Quantitative characteristics were compared using student’s *t*-test and Mann-Whitney U test. Average annual percentage changes (AAPC) were also calculated for the whole time period (2009–2016). *p*-value < 0.05 was considered statistically significant.

## 3. Results

### 3.1. Sample Characteristics

A total of 92373 people were examined. The study sample consisted of 41.6 % (*n = 38,412*) men aged 40–54 and 58.4 % (*n = 53,961*) women aged 50–64 without overt cardiovascular disease. The mean (SD) age of the whole study sample was 52.15 (±6.21): 55.85 (±4.40) among women and 46.96 (±4.39) among men. Most people surveyed among men and women were in the subgroup aged 50–54 years (14.3% (*n = 13,249*) of men and 26.0% (*n = 23,995*) of women), also there was 14.0% (*n = 12,923*) aged 40–44 and 13.3% (*n = 12,240*) aged 45–49 in a group of male subjects, and 17.7% (*n = 16,348*) aged 55–59 and 14.7% (*n = 13,618*) aged 60–64 in a group of women.

As long as LitHiR primary prevention program includes measurements of other frequent modifiable risk factors of cardiovascular disease, we compared them among daily smokers and non-smokers (Table 1).

### 3.2. Trends in Prevalence of Smoking

There was a statistically significant upward trend in the number of smoking subjects (average annual percent change (AAPC) 2.99%; *p* < 0.001). When comparing study samples‘ groups by gender, the prevalence of smoking among men remained quite constant yet significantly decreased (AAPC −0.73%, *p* = 0.02) while the prevalence of smoking women showed a more than 1,5-fold increase (AAPC 6.89%, *p* < 0.001) (Figure 1).

Smoking prevalence among females was the highest in the group aged 50–54 and the number of daily smokers in this group significantly increased during the 2009–2016 period (AAPC 4.96%, *p* < 0.001). The same statistically significant upward trend was found among females aged 55–59 (AAPC 7.13%, *p* < 0.001) and 60–64 years (AAPC 8.80%, *p* < 0.001) (Figure 2).

Smoking prevalence in a male group aged 40–44 significantly decreased (AAPC −3.87%, *p* < 0.001) while there was no significant change in another two male groups aged 45–49 and 50–54 years (*p* = 0.139 and *p* = 0.166 respectively) (Figure 3).

### 3.3. Trends in Smoking Cessation Activity

Despite the fact that the prevalence of smoking has increased, the study also showed a significant increase in the percentage of smoking quitters in the whole group (AAPC 7.22%, *p* < 0.001). The same tendency was noticed in groups of men and women separately (AAPC 7.22%, *p* = 0.006 and AAPC 11.73%, *p* < 0.001 respectively) (Figure 4).

### 3.4. Trends in Smoking Intensity

The study showed that mostly current smokers smoke 10–20 cigarettes per day in the whole group although the number has significantly decreased from 71.90% in 2009 to 69.60% in 2016. The percentage of smokers who smoke <10 cigarettes per day significantly increased from 23.80% in 2009 to 26.50% in 2016 (*p* = 0.002). On the other hand, there was no significant change of smoking intensity in groups of male and female smokers separately (*p* = 0.221 and *p* = 0.113 respectively). Male smokers mostly smoke 10–20 cigarettes per day (79.40% in 2009 and 77.80% in 2016) while half of smoking females mostly smoke <10 cigarettes per day (50.60% in 2009 and 51.10% in 2016) and almost half of them 10–20 cigarettes per day (48.80% in 2009 and 48.10% in 2016). There was no significant change in smoking intensity in all age groups.

## 4. Discussion

The global age-standardized prevalence of smoking has decreased steadily since the beginning of the 21^st^ century (from 26.9% in 2000 to 20.2% in 2015), especially in high-income countries (e.g., Great Britain, Australia, the United States, Finland and Japan). According to WHO, smoking rates appear to be decreasing in almost all regions except African and East Mediterranean, although the number of smokers globally is still very large (1114 million of smokers aged ≥15 years in 2015) [16]. The European region still has the highest average of the number of daily smokers (28%) [10], while smoking prevalence among 28 European Union countries is 18.4% [11]. The difference could be explained by the fact that the WHO European region also includes countries with one of the highest smoking rates like Russian Federation, Turkey and Ukraine [10,11]. The highest smoking rates in EU were estimated in Bulgaria (27.3%) and Greece (27.0%) while the lowest prevalence of smoking is found in Nordic countries—Sweden (8.7%), Finland (11.6%) and Denmark (12.3%) [11].

In comparison with other WHO regions, the main difference of smoking prevalence in Europe is that the number of female smokers is much higher making the gap between the prevalence of smoking men and women very small, especially in countries such as Austria, Denmark, Ireland, the Netherlands and the UK [17]. In the latest data of Eurostat, the gender difference of smoking prevalence was estimated in percentage points and showed that countries with higher smoking rates had a bigger gender difference—there was a 24.4% in Lithuania, 15.5 % in Bulgaria and 12.0% in Greece between the prevalence rates of male and female smokers. There was a completely different situation in Finland (2.2%), Denmark (0.7%) and Sweden (−2.3%) showing that smoking prevalence among women was almost the same or even higher compared with men [11].

In 1993–1994, international tobacco companies found out that the number of smoking women in Lithuania was very low—just about 6%, so considerable amounts of tobacco advertisements were directed towards women [18]. The obtained data from Finbalt Health Monitor project showed that smoking prevalence increased among both gender groups until 2000, with a significant rise of 2.3 times among women smokers. Since 2000, the prevalence of male smokers started decreasing while the rate of smoking women further remained quite constant [19].

A downtrend in smoking prevalence could be related to active positive policy developments in Lithuania since the Law on Tobacco Control of the Republic of Lithuania was adopted in 1996 and then followed with a ban on advertising tobacco in 2000 [20]. Due to smoking being the single greatest cause of preventable death in 2003, several countries, including the EU as a whole and its member states, have agreed on reduction targets for smoking—the WHO Framework Convention on Tobacco Control (WHO FCTC). It suggests that six cost-effective MPOWER measures should be in a comprehensive tobacco control program of each country including monitoring, smoke-free policies, cessation, health warnings, advertising bans and taxation. Lithuania regards WHO FCTC recomendations and periodically delivers recent, representative data of smoking among adults and youth; bans smoking in public places like bus stops and cafes (since 2007); puts health warnings on tobacco packages with all appropriate characteristics; bans promoting or advertising tobacco on national television, radio and print media; and has made sure that more than 75% of tobacco retail price is tax (tax increases in 2004, 2007–2010, 2012–2015) [21,22]. Also, on 1 January 2017, Lithuania’s Law banned tobacco producers from sponsoring events or using mass media to promote the use of tobacco, and on 26 January 2017, undertook a comprehensive ban on any form of promoting to purchase or consume any tobacco products [23].

Despite all governmental laws, smoking remains a big problem in Lithuania with a much higher smoking rate than the EU average [6], so all measures to reduce it seem to be insufficient. The most effective measure to reduce smoking rates appeared to be increasing taxation [24]. According to TCS data, Lithuania got only 12 points out of 30 on price increasing [25]. Taxation could be mostly affected by the illegal supply of cigarettes through the external borders from neighbouring countries, which additionally puts pressure on tax levels there. Although there are no official estimates on illicit cigarettes in Lithuania, there are some reasons why it became a major problem there. First of all, Lithuania has an external European Union border with Belarus and Russia (Kaliningrad) and is close to Ukraine, them being the main sources of illegal tobacco. This specific geography of Lithuania is an important gateway for tobacco trade to other EU Member states. Also, there is a huge taxation issue—in Belarus and Russia retail prices of cigarettes are up to fives times lower than in Lithuania. Secondly, smuggling has deep roots in Lithuania and gets a big social acceptance. Cigarettes are top amongst the most popular illegal products [26]. Experiencing a high rate of illicit trade in cigarettes, Lithuania ratified The Protocol to Eliminate Illicit Trade in Tobacco products in 2016 to stop a serious threat to public health from neighbouring countries [27].

## 5. Conclusions

Despite all the international commitments and governmental efforts to reduce smoking (bans on promoting or advertising tobacco, taxation, health warnings on packages, bans on smoking in most public places, efforts to reduce illicit trade etc), the analysis of primary prevention program’s results showed that smoking prevalence among men aged 40–54 and women aged 50–64 increased due to a 1,5-fold increase in the number of female smokers during the 2009–2016 period. On the other hand, the study also showed a significant increase in the percentage of smoking quitters in the whole group.

## Figures and Tables

**Figure 1 medicina-55-00130-f001:**
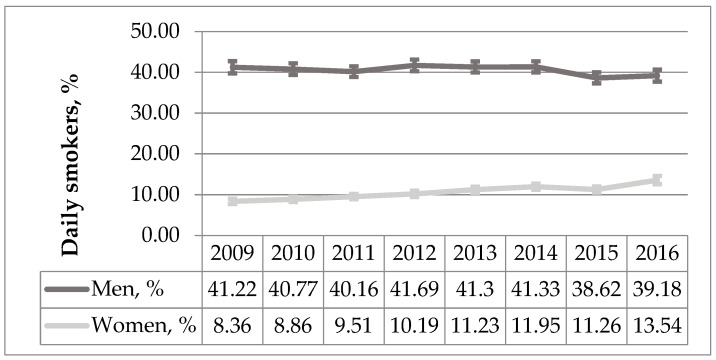
The percentage of daily smokers in gender groups during 2009–2016 period (%, 95% CI).

**Figure 2 medicina-55-00130-f002:**
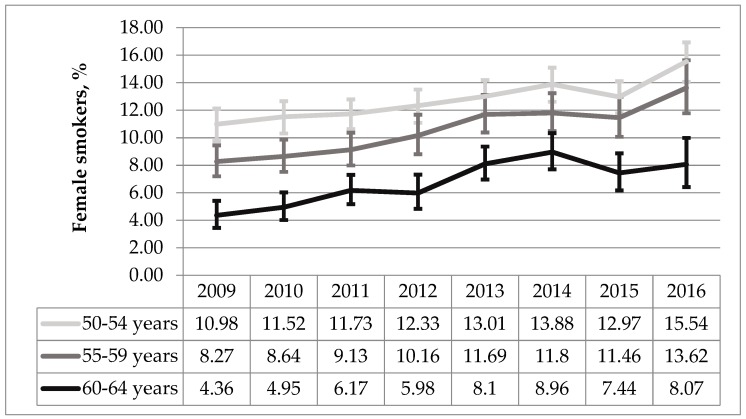
The trend analysis of female smokers in age groups during 2009–2016 period (%, 95% CI).

**Figure 3 medicina-55-00130-f003:**
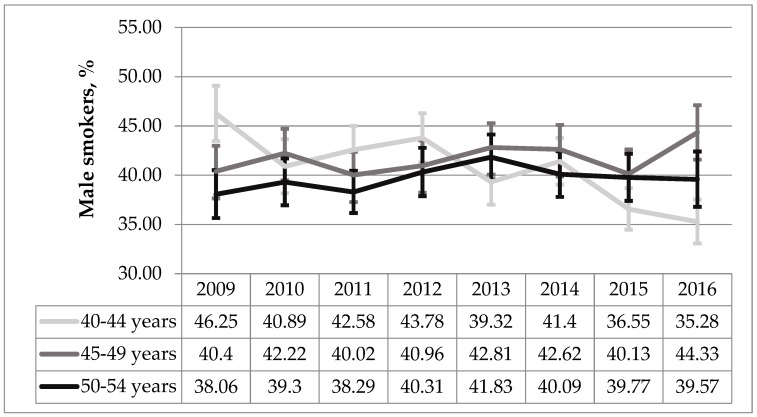
The trend analysis of male smokers in age groups during 2009–2016 period (%, 95% CI).

**Figure 4 medicina-55-00130-f004:**
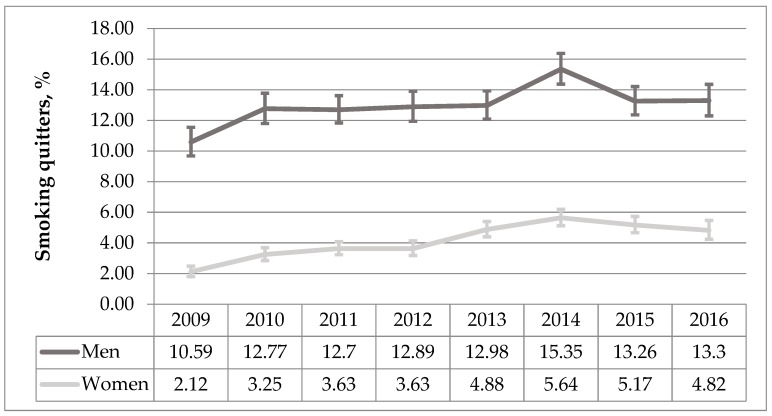
The trend analysis of smoking cessation activity among men and women during 2009–2016 period (%, 95% CI).

**Table 1 medicina-55-00130-t001:** Prevalence of the main modifiable cardiovascular risk factors among smokers and non-smokers, % (95% CI).

Risk Factor	Smoking Men		Non-Smoking Men	
	Frequency	% (95% CI)	Frequency	% (95% CI)
Arterial hypertension	7227	46.43 (45.65–47.22)	11188	48.97 (48.32–49.62)
Dyslipidemia	13469	86.54 (85.99–87.07)	19934	87.25 (86.81–87.68)
Diabetes mellitus	1574	10.11 (9.64–10.6)	2365	10.35 (9.96–10.75)
Obesity	3803	24.43 (23.76–25.12)	7050	30.86 (30.26–31.46)
	**Smoking Women**		**Non-Smoking Women**	
Arterial hypertension	3057	54.07 (52.76–55.37)	28845	59.71 (59.27–60.15)
Dyslipidemia	5234	92.57 (91.86–93.24)	44256	91.61 (91.36–91.86)
Diabetes mellitus	622	11 (10.2–11.85)	5336	11.05 (10.77–11.33)
Obesity	2690	47.58 (46.27–48.89)	26865	55.61 (55.17–56.06)

CI—confidence interval.

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
