# Peer review of "Trends in Cigarette Smoking among Middle-Aged Lithuanian Subjects Participating in the Primary Prevention Program between 2009 and 2016"

_medicina, 2019, doi:10.3390/medicina55050130_

Reviewer 1 Report

Introduction:

Lines 30-47 These are a bit muddled.  I would mention all the global stuff and then move to Europe and then Lithuania

The risk of some illnesses remains  high for longer after quitting smoking

Line 49 Smoking may also cause high blood pressure and diabetes so this is a bit confusing

Table 1  Can you separate by smokers and non smokers (by gender) and say if there are significant differences

Figure 1.  Can you include confidence intervals and the proportion of quitters

Please present the age stratified results in a table or figure.  I am a bit confused as to why the prevalence is growing in women if the highest prevalence is aged ~50 years?

Have tobacco companies been marketing at women?  Do women have more disposable income than previously?

 Are there ethnic differences e.g. do ethnic Russian smoke more than ethnic Lithuanians if the smuggled cigarettes are coming from Russia?

I would expect the ‘Tobacco Epidemic’ to be referenced

 How well are the smoke free laws upheld in Lithuania?

 I think you need to make more of women in a EU country are smoking more

Author Response

We appreciate the reviewer’s comments. The following are our point-by-point responses:

Comment #1: Lines 30-47 These are a bit muddled. I would mention all the global stuff and then move to Europe and then Lithuania. Line 49 Smoking may also cause high blood pressure and diabetes so this is a bit confusing.

Response: We improved ‘Introduction’ section by changing the order of sentences so that global, then Europe’s and then Lithuania’s situation of deaths from CVD would be logically one after another. We more clearly specified in the text that we took smoking just one as CVD risk factors.

Comment #2: Table 1  Can you separate by smokers and non smokers (by gender) and say if there are significant differences. Figure 1.  Can you include confidence intervals and the proportion of quitters.

Response: We added this additional data. You can find a new table with separated other CVD risk factors between smokers and non smokers and also by gender. We also included confidence intervals in all graphs and made a figure representing the proportion of quitters.

Comment #3: Please present the age stratified results in a table or figure.  I am a bit confused as to why the prevalence is growing in women if the highest prevalence is aged ~50 years?

Response: We added some additional figures with age stratified results.

Comment #4: Have tobacco companies been marketing at women?  Do women have more disposable income than previously? Are there ethnic differences e.g. do ethnic Russian smoke more than ethnic Lithuanians if the smuggled cigarettes are coming from Russia? How well are the smoke free laws upheld in Lithuania?

Response: Regarding your comments, we improved the whole ‘Discussion’ section and added additional information.

Reviewer 2 Report

Comments: Smoking Trends in Lithuanian Middle-aged Subjects Participating in the Primary Prevention Program Between 2009 and 2016

 Thank you for the manuscript on important data from Lithuania.  The sample size and scope of the LitHiR are well described and lend strength to the manuscript.  My overall feedback and specific comments follow.

 Overall

the objective of the manuscript is to examine the trends in smoking by age and gender.  The introduction is well articulated in the relationship between tobacco use and chronic disease conditions.  However, the focus is lost after the introduction as the manuscript is about tobacco use and trends over time in that indicator for certain population groups. It would be good to focus on the rationale a bit more (i.e., older adults are more likely to have chronic conditions and tobacco use is one preventive measure….etc.).  

In general, a good edit is needed for the entire manuscript to be clear on what is being communicated.  The sample characteristics shown in table 1 represent the medical profile of the population.  While these are important, it might be worth telling the reader what they are indicating in terms of your study objective.

 One question I had was about the analytic methods—these data can be fully examined using Trend analyses and some way to look at the stratification by age and gender.  The differences and the trajectory of change is quite different for male than females.

 Discussion section needs to rewritten.

 Specific comments

Introduction: nicely done

Methods: given the sample sizes indicated in lines 70-75, I think you can actually look at trends over time.  While the description of chronic disease indicators is interesting—how does it relate to your analysis?  Since you did not look at the chronic indicators as outcomes, please justify their inclusion or take it out.

 Given your sample size, not sure you need to go out to the second digit after the decimal point.  It might help to simply the presentation a little.  Also, if you decide to conduct multivariable analyses—consider age, gender interaction as data for some your analyses indicate this line of inquiry.

Measure of smoking—what is it measuring—cigarettes or all smoked products (i.e., hooka, HNB)?  Clarify what went into your measurement and that also applies to definition of quitting (for how long)?

 Table 1.  Since age is one of your major indicators of interest, I wondered why it was not included.  Also, not sure how this table relates to your research objective.  Either indicate its relevance or potentially consider taking it out.

Lines 103-107—It would be better if you analyzed the whole dataset using trend analysis.  Presenting only 2009 and 2016 end years—at a minimum include relative change and whether that is statistically significant or not.

 Figure 1.  Maybe better as a line graph where x-axis is year and y-axis are prevalence estimates.  The title of this figure needs to be corrected in that “percent” is presented and not numbers.  Please revise.

 Lines 119-127—this is an important indicator and your data showing an interesting, however, slow progression—graph might be useful as well.

 Lines 119-127—your data show that smoking increased among women and stayed pretty constant for men—

 Lines 132-133-please clarify what you mean here-

 Discussion beginning on line 141 needs to be restructured and rewritten. No need to repeat your results (lines 152-156).  Suggest, a brief summary of what you found and interpretation of your findings within the context of other research, followed by the context-geographic, and then what actions/recommendations are needed with the FCTC-MPOWER framework. 

Lines 181-183—this text appears to indicate that you are saying that the gov. of Sweden encouraged switching from combustible to smokeless “snus” use—Did you mean that or does the reference you cite indicate that in some way?   The observed phenomenon may have resulted from the policies enacted; however, one has to be very careful about making attribution—please clarify what you mean.

 Conclusion, lines 201-205—your data only refer to Lithuania—so you may want to be specific in saying that despite the efforts X,Y, Z in Lithuania to be sure that you are referring to national efforts and their potential impact.

This is also your opportunity to present a recommendation or two to help the potential readers to see how the results can be used to make impactful change.

Author Response

We appreciate the reviewer’s comments. Regarding to an overall comment we improved our manuscript. The following are our point-by-point responses:

Comment #1: While the description of chronic disease indicators is interesting—how does it relate to your analysis?  Since you did not look at the chronic indicators as outcomes, please justify their inclusion or take it out.

Response: We decided to delete this paragraph.

Comment #2: Measure of smoking—what is it measuring—cigarettes or all smoked products (i.e., hooka, HNB)?  Clarify what went into your measurement and that also applies to definition of quitting (for how long)?

Response: We clarified this information in the text.

Comment #3: Table 1.  Since age is one of your major indicators of interest, I wondered why it was not included.  Also, not sure how this table relates to your research objective.  Either indicate its relevance or potentially consider taking it out.

Response: We improved the old table with new additional data (also Table 1) showing frequency of other CVD risk factors among smokers and non-smokers.

Comment #4: Lines 103-107—It would be better if you analyzed the whole dataset using trend analysis.  Presenting only 2009 and 2016 end years—at a minimum include relative change and whether that is statistically significant or not.

Response: We improved our data analysis in graphs and added relative change rate (average annual percentage change) talking about results.

Comment #5: Figure 1.  Maybe better as a line graph where x-axis is year and y-axis are prevalence estimates.  The title of this figure needs to be corrected in that “percent” is presented and not numbers.  Please revise. Lines 119-127—this is an important indicator and your data showing an interesting, however, slow progression—graph might be useful as well. Lines 119-127—your data show that smoking increased among women and stayed pretty constant for men—. Lines 132-133-please clarify what you mean here-

Response: We revised and improved Figure 1. Also added additional figures.

Comment #6: Discussion beginning on line 141 needs to be restructured and rewritten. No need to repeat your results (lines 152-156).  Suggest, a brief summary of what you found and interpretation of your findings within the context of other research, followed by the context-geographic, and then what actions/recommendations are needed with the FCTC-MPOWER framework.

Response: The whole ‘Discussion’ section was rewritted. Regarding to your suggestions, we improved this part.

Comment #7: Lines 181-183—this text appears to indicate that you are saying that the gov. of Sweden encouraged switching from combustible to smokeless “snus” use—Did you mean that or does the reference you cite indicate that in some way?   The observed phenomenon may have resulted from the policies enacted; however, one has to be very careful about making attribution—please clarify what you mean.

Response: We decided to take this information out.

Comment #8: Conclusion, lines 201-205—your data only refer to Lithuania—so you may want to be specific in saying that despite the efforts X,Y, Z in Lithuania to be sure that you are referring to national efforts and their potential impact.

Response: We improved our ‘Conclusions’ section.

Round  2

Reviewer 1 Report

This paper looks at smoking prevalence in Lithuania by gender.  I have spotted some potential issues with the data analysis.

Line 33 “even though” should this be deleted?

Data: what is the response rate for the survey?  Is it clustered or a random sample?  If clustered this would need to be taken into account in the analysis

Sample characteristics: are they similar to the Lithuanian population?  If not then your bivariable analysis needs to be weighted

Author Response

Dear reviewers, We appreciate the reviewer’s comments. The following are our point-by-point responses: Comment #1: Line 33 “even though” should this be deleted? Response: We deleted this part. Comment #2: Data: what is the response rate for the survey? Is it clustered or a random sample? If clustered this would need to be taken into account in the analysis. Response: We specified the process of random sampling of this study in ‘Methods’ section. Comment #3: Sample characteristics: are they similar to the Lithuanian population? If not then your bivariable analysis needs to be weighted. Response: We think that LitHiR primary prevention programme data represents the whole middle-aged Lithuanian population, because 94.8% (398/420) of all the primary care institutions participate in the project, which uniformly cover the whole country. This report describes the trends in smoking in the randomly selected group of 92373 subjects involved in the LitHiR programme during the period 2009-2016 at the primary level. Number of people examined in the primary prevention programme is presented in figure 1. In 2016, 256625 adults were examined in primary care centers, covering about 37.5% of all target population. Detailed description of the LitHiR programme protocol is provided in reference [15] of the manuscipt. Manuscript updated according to the comment (you can find it in ‘Methods’ section). Sincerely, Dr. Egidija Rinkūnienė
